# AN ADVERSARIAL LEARNING FRAMEWORK FOR A PERSONA-BASED MULTI-TURN DIALOGUE MODEL

## ABSTRACT

In this paper, we extend the persona-based sequence-to-sequence (Seq2Seq) neural network conversation model to a multi-turn dialogue scenario by modifying the state-of-the-art hredGAN architecture to simultaneously capture utterance attributes such as speaker identity, dialogue topic, speaker sentiments and so on. The proposed system, phredGAN has a persona-based HRED generator (PHRED) and a conditional discriminator. We also explore two approaches to accomplish the conditional discriminator: (1) $phredGAN_a$, a system that passes the attribute representation as an additional input into a traditional adversarial discriminator, and (2) $phredGAN_d$, a dual discriminator system which in addition to the adversarial discriminator, collaboratively predicts the attribute(s) that generated the input utterance. To demonstrate the superior performance of phredGAN over the persona SeqSeq model, we experiment with two conversational datasets, the Ubuntu Dialogue Corpus (UDC) and TV series transcripts from the Big Bang Theory and Friends. Performance comparison is made with respect to a variety of quantitative measures as well as crowd-sourced human evaluation. We also explore the trade-offs from using either variant of $phredGAN$ on datasets with many but weak attribute modalities (such as with Big Bang Theory and Friends) and ones with few but strong attribute modalities (customer-agent interactions in Ubuntu dataset).

## 1 INTRODUCTION

Recent advances in machine learning especially with deep neural networks has lead to tremendous progress in natural language processing and dialogue modeling research (Sutskever et al., 2014; Vinyals & Le, 2015; Serban et al., 2016). Nevertheless, developing a good conversation model capable of fluent interaction between a human and a machine is still in its infancy stage. Most existing work relies on limited dialogue history to produce response with the assumption that the model parameters will capture all the modalities within a dataset. However, this is not true as dialogue corpora tend to be strongly multi-modal and practical neural network models find it difficult to disambiguate characteristics such as speaker personality, location and sub-topic in the data.

Most work in this domain has primarily focused on optimizing dialogue consistency. For example, Serban et al. (Serban et al., 2016; 2017b;a) and Xing et al. (2017) introduced a Hierarchical Recurrent Encoder-Decoder (HRED) network architecture that combines a series of recurrent neural networks to capture long-term context state within a dialogue. However, the HRED system suffers from lack of diversity and does not have any guarantee on the generator output since the output conditional probability is not calibrated. Olabiyi et al. (2018) tackles these problems by training a modified HRED generator alongside an adversarial discriminator in order to increase diversity and provide a strong and calibrated guarantee to the generator's output. While the hredGAN system improves upon response quality, it does not capture speaker and other attributes modality within a dataset and fails to generate persona specific responses in datasets with multiple modalities.

On the other hand, there has been some recent work on introducing persona into dialogue models. For example, Li et al. (2016b) integrates attribute embeddings into a single turn (Seq2Seq) generative dialogue model. In this work, Li et al. consider persona models one with Speaker-only representation and the other with Speaker and Addressee representations (Speaker-Addressee model), both of which capture certain speaker identity and interactions. Nguyen et al. (2018) continue along the

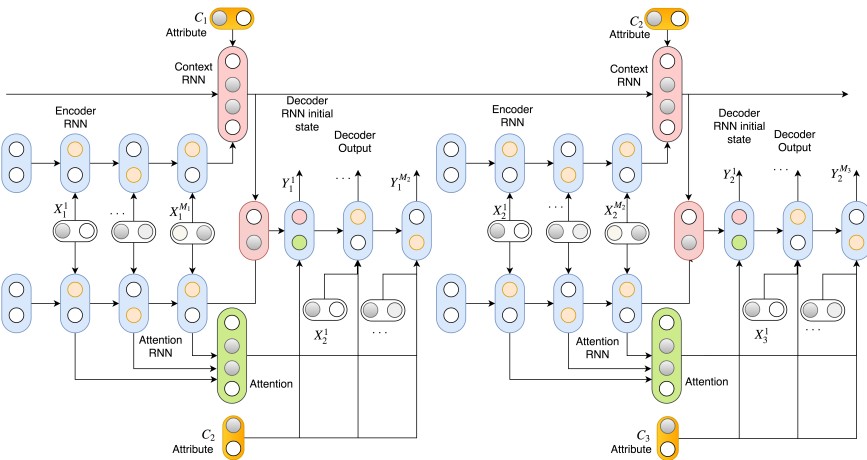

Figure 1: **The PHRED generator with local attention -** The attributes C, allows the generator to condition its response on the utterance attributes such as speaker identity, subtopics and so on.

same line of thought by considering a Seq2Seq dialogue model with Responder-only representation. In both of these cases, the attribute representation is learned during the system training. Zhang et al. (2018) proposed a slightly different approach. Here, the attributes are a set of sentences describing the profile of the speaker. In this case, the attributes representation is not learned. The system however learns how to attend to different parts of the attributes during training. Still, the above persona-based models have limited dialogue history (single turn); suffer from exposure bias worsening the trade-off between personalization and conversation quality and cannot generate multiple responses given a dialogue context. This is evident in the relatively short and generic responses produced by these systems, even though they generally capture the persona of the speaker.

In order to overcome these limitations, we propose two variants of an adversarially trained persona conversational generative system, $phredGAN$, namely $phredGAN_a$ and $phredGAN_d$. Both systems aim to maintain the response quality of $hredGAN$ and still capture speaker and other attribute modalities within the conversation. In fact, both systems use the same generator architecture (PHRED generator), i.e., an $hredGAN$ generator (Olabiyi et al., 2018) with additional utterance attribute representation at its encoder and decoder inputs as depicted in Figure 1. Conditioning on external attributes can be seen as another input modality as is the utterance into the underlying system. The attribute representation is an embedding that is learned together with the rest of model parameters similar to Li et al. (2016b). Injecting attributes into a multi-turn dialogue system allows the model to generate responses conditioned on particular attribute(s) across conversation turns. Since the attributes are discrete, it also allows for exploring different what-if scenarios of model responses. The difference between the two systems is in the discriminator architecture based on how the attribute is treated.

We train and sample both variants of $phredGAN$ similar to the procedure for $hredGAN$ (Olabiyi et al., 2018). To demonstrate model capability, we train on a customer service related data such as the Ubuntu Dialogue Corpus (UDC) that is strongly bimodal between question poser and answerer, and transcripts from a multi-modal TV series *The Big Bang Theory* and *Friends* with quantitative and qualitative analysis. We examine the trade-offs between using either system in bi-modal or multi-modal datasets, and demonstrate system superiority over state-of-the-art persona conversational models in terms of dialogue response quality and quantitatively with perplexity, BLEU, ROUGE and distinct n-gram scores.

## 2   MODEL ARCHITECTURE

In this section, we briefly introduce the state-of-the-art $hredGAN$ model and subsequently show how we derive the two persona versions by combining it with the distributed representation of the dialogue speaker and utterance attributes, or with an attribute discrimination layer at the end of the model pipeline.

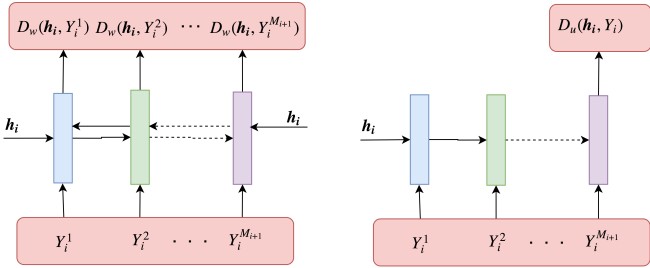

Figure 2: **The** $phredGAN_d$ **dual discriminator - Left:** $D_{adv}$ is a word-level discriminator used by both $phredGAN_a$ and $phredGAN_d$ to judge normal dialogue coherency as in $hredGAN$. **Right:** $D_{att}$, an utterance-level attribute discriminator is used only in $phredGAN_d$ to predict the likelihood a given utterance was generated from a particular attribute.

## 2.1 $hredGAN$: ADVERSARIAL LEARNING FRAMEWORK

**Problem Formulation**: The $hredGAN$ (Olabiyi et al., 2018) formulates multi-turn dialogue response generation as: given a dialogue history of sequence of utterances, $\boldsymbol{X_i} = (X_1, X_2, \cdots, X_i)$, where each utterance $X_i = (X_i^1, X_i^2, \cdots, X_i^{M_i})$ contains a variable-length sequence of $M_i$ word tokens such that $X_i^j \in V$ for vocabulary $V$, the dialogue model produces an output $Y_i = (Y_i^1, Y_i^2, \cdots, Y_i^{T_i})$, where $T_i$ is the number of generated tokens. The framework uses conditional GAN structure to learn a mapping from an observed dialogue history to a sequence of output tokens. The generator, $G$, is trained to produce sequences that cannot be distinguished from the ground truth by an adversarially trained discriminator, $D$ akin to a two-player min-max optimization problem. The generator is also trained to minimize the cross-entropy loss $\mathcal{L}_{MLE}(G)$ between the ground truth $X_{i+1}$, and the generator output $Y_i$. The following objective summarizes both goals:

$$G^*, D^* = arg \min_G \max_D \left( \lambda_G \mathcal{L}_{cGAN}(G, D) + \lambda_M \mathcal{L}_{MLE}(G) \right). \tag{1}$$

where $\lambda_G$ and $\lambda_M$ are training hyperparamters and $\mathcal{L}_{cGAN}(G, D)$ and $\mathcal{L}_{MLE}(G)$ are defined in Eqs. (5) and (7) of Olabiyi et al. (2018) respectively. Please note that the generator $G$ and discriminator $D$ share the same encoder and embedding representation of the word tokens.

## 2.2 $phredGAN$: PERSONA ADVERSARIAL LEARNING FRAMEWORK

The proposed architecture of $phredGAN$ is very similar to that of $hredGAN$ (Olabiyi et al., 2018). The only difference is that the dialogue history is now $\boldsymbol{X_i} = ((X_1, C_1), (X_2, C_2), \cdots, (X_i, C_i))$ where $C_i$ is additional input that represents the speaker and/or utterance attributes. Please note that $C_i$ can either be a sequence of tokens or single token such that $C_i^j \in Vc$ for vocabulary $Vc$. Also, at the $ith$ turn, $C_i$ and $C_{i+1}$ are the source/input attribute and target/output attribute to the generator respectively. The embedding for attribute tokens is also learned similar to that of word tokens.

Both versions of $phredGAN$ shares the same generator architecture (PHRED) but different discriminators. Below is the highlight of how they are derived from the $hredGAN$ architecture.

**Encoder**: The context RNN, $cRNN$ takes the source attribute $C_i$ as an additional input by concatenating its representation with the output of $eRNN$ as in Figure 1. If the attribute $C_i$ is a sequence of tokens, then an attention (using the output of $eRNN$) over the source attribute representations is concatenated with the output of $eRNN$. This output is used by the generator to create a context state for a turn $i$.

**Generator**: The generator decoder RNN, $dRNN$ takes the target attribute $C_{i+1}$ as an additional input as in Fig. 1. If the attribute $C_{i+1}$ is a sequence of tokens, then an attention (using the output of $dRNN$) over the attribute representations is concatenated with the rest of the decoder inputs. This forces the generator to draw a connection between the generated responses and the utterance attributes such as speaker identity.

**Noise Injection**: As in Olabiyi et al. (2018), we also explore different noise injection methods.

**Objective:** For $phredGAN$, the optimization objective in eq. (1) can be updated as:

$$G^*, D^*_{adv}, D^*_{att} = arg \min_G \big( \max_{D_{adv}} \lambda_{G_{adv}} \mathcal{L}^{adv}_{cGAN}(G, D_{adv}) \tag{2}$$
$$+ \min_{D_{att}} \lambda_{G_{att}} \mathcal{L}^{att}_c(G, D_{att}) + \lambda_M \mathcal{L}_{MLE}(G) \big).$$

where $\mathcal{L}^{adv}_{cGAN}(G, D_{adv})$ and $\mathcal{L}^{att}_c(G, D_{att})$ are the traditional adversarial and attribute prediction loss respectively and dependent on the architectural variation. It is worth to point out that while the former is adversarial, the later is collaborative in nature. The MLE loss is common and can be expressed as:

$$\mathcal{L}_{MLE}(G) = \mathbb{E}_{X_{i+1}}[-log\, P_G(X_{i+1}|\boldsymbol{X_i}, C_{i+1}, Z_i)]. \tag{3}$$

where $Z_i$ the noise sample and depends on the choice of either utterance-level or word-level noise input into the generator (Olabiyi et al., 2018).

### 2.3 $phredGAN_a$: ATTRIBUTES AS A DISCRIMINATOR INPUT

$phredGAN_a$ shares the same discriminator architecture as the $hredGAN$ but with additional input, $C_{i+1}$. Since it does not use attribute prediction, $\lambda_{G_{att}} = 0$.

The adversarial loss, $\mathcal{L}^{adv}_{cGAN}(G, D)$ can then be expressed as:

$$\mathcal{L}^{adv}_{cGAN}(G, D_{adv}) = \mathbb{E}_{\boldsymbol{X_i}, C_{i+1}, X_{i+1}}[\log D_{adv}(\boldsymbol{X_i}, C_{i+1}, X_{i+1})] \tag{4}$$
$$+ \mathbb{E}_{\boldsymbol{X_i}, C_{i+1}, Z_i}[1 - \log D_{adv}(\boldsymbol{X_i}, C_{i+1}, G(\boldsymbol{X_i}, C_{i+1}, Z_i))]$$

The addition of speaker or utterance attributes allows the dialogue model to exhibit personality traits given consistent responses across style, gender, location, and so on.

### 2.4 $phredGAN_d$: ATTRIBUTES AS A DISCRIMINATOR TARGET

$phredGAN_d$ does not take the attribute representation at its input but rather use the attributes as the target of an additional discriminator $D_{att}$. The adversarial and the attribute prediction losses can be respectively expressed as:

$$\mathcal{L}^{adv}_{cGAN}(G, D_{adv}) = \mathbb{E}_{\boldsymbol{X_i}, X_{i+1}}[\log D_{adv}(\boldsymbol{X_i}, X_{i+1})] \tag{5}$$
$$+ \mathbb{E}_{\boldsymbol{X_i}, Z_i}[1 - \log D_{adv}(\boldsymbol{X_i}, G(\boldsymbol{X_i}, C_{i+1}, Z_i))]$$

$$\mathcal{L}^{att}_c(G, D_{att}) = \mathbb{E}_{C_{i+1}}[-\log D_{att}(C_{i+1}|\boldsymbol{X_i}, X_{i+1})] \tag{6}$$
$$+ \mathbb{E}_{C_{i+1}}[-\log D_{att}(C_{i+1}|\boldsymbol{X_i}, G(\boldsymbol{X_i}, C_{i+1}, Z_i))]$$

**Attribute Discriminator**: In addition to the existing word-level adversarial discriminator $D_{adv}$ from $hredGAN$, we add an attribute discriminator, $D_{att}$, that discriminates on an utterance level to capture attribute modalities since attributes are assigned at utterance level. The discriminator uses a unidirectional RNN ($D_{attRNN}$) that maps the input utterance to the particular attribute(s) that generated it. The attributes can be seen as hidden states that inform or shape the generator outputs. The attribute discriminator can be expressed as:

$$D_{att}(C_{i+1}|\boldsymbol{X_i}, \chi) = D_{attRNN}(\boldsymbol{h_i}, E(\chi)) \tag{7}$$

where $E(.)$ is the word embedding lookup (Olabiyi et al., 2018), $\chi = X_{i+1}$ for groundtruth and $\chi = Y_i$ for the generator output.

## 3 MODEL TRAINING AND INFERENCE

### 3.1 MODEL TRAINING

We train both the generator and the discriminator (with shared encoder) of both variants of $phredGAN$ using the training procedure in Algorithm 1 (Olabiyi et al., 2018). For both variants, $\lambda_{G_{adv}} = \lambda_M = 1$, and for $phredGAN_a$ and $phredGAN_d$, $\lambda_{G_{att}} = 0$ and $\lambda_{G_{att}} = 1$ respectively. Since the encoder, word embedding and attribute embedding are shared, we are able to train the system end-to-end with back-propagation.

**Encoder**: The encoder RNN, $eRNN$, is bidirectional while $cRRN$ is unidirectional. All RNN units are 3-layer GRU cell with hidden state size of 512. We use word vocabulary size, $V = 50,000$ with word embedding size of 512. The number of attributes, $Vc$ is dataset dependent but we use an attribute embedding size of 512. In this study, we only use one attribute per utterance so that is no need to use attention to combine the attribute embeddings.

**Generator**: The generator decoder RNN, $dRNN$ is also a 3-layer GRU cell with hidden state size of 512. The $aRNN$ outputs are connected to the $dRNN$ input using an additive attention mechanism (Bahdanau et al., 2015).

**Adversarial Discriminator**: The word-level discriminator RNN, $D_{RNN}$ is a bidirectional RNN, each 3-layer GRU cell with hidden state size of 512. The output of both the forward and the backward cells for each word are concatenated and passed to a fully-connected layer with binary output. The output is the probability that the word is from the ground truth given the past and future words of the sequence, and in the case of $phredGAN_a$, the responding speaker's embedding.

**Attribute Discriminator**: The attribute discriminator RNN, $D_{attRNN}$ is a unidirectional RNN with a 3-layer GRU cell, each of hidden state size 512. A softmax layer is then applied to project the final hidden state to a prespecified number of attributes, $V_c$. The output is the probability distribution over the attributes.

**Others**: All parameters are initialized with Xavier uniform random initialization (Glorot & Bengio, 2010). Due to the large word vocabulary size, we use sampled softmax loss (Jean et al., 2015) for MLE loss to expedite the training process. However, we use full softmax for model evaluation. For both systems, parameters updates are conditioned on the word-level discriminator accuracy performance as in Olabiyi et al. (2018) with $acc_{D_{adv}^{th}} = 0.99$ and $acc_{G_{th}} = 0.75$. The model is trained end-to-end using the stochastic gradient descent algorithm. Finally, the model is implemented, trained, and evaluated using the TensorFlow deep learning framework.

## 3.2 Model Inference

We use an inference strategy similar to the approach in Olabiyi et al. (2018).

For the modified noise sample, we perform a linear search for $\alpha$ with sample size $L = 1$ based on the average word-level discriminator loss, $-logD_{adv}(G(.))$ (Olabiyi et al., 2018) using trained models run in autoregressive mode to reflect performance in actual deployment. The optimum $\alpha$ value is then used for all inferences and evaluations. During inference, we condition the dialogue response generation on the encoder outputs, noise samples, word embedding and the attribute embedding of the intended responder. With multiple noise samples, $L = 64$, we rank the generator outputs by the discriminator which is also conditioned on encoder outputs, and the intended responder's attribute embedding. The final response is the response ranked highest by the discriminator. For $phredGAN_d$, we average the confidences produced by $D_{adv}$ and $D_{att}$.

## 4 Experiments and Results

In this section, we explore the performance of PHRED, $phredGAN_a$ and $phredGAN_d$ on two conversational datasets and compare its performance to non-adversarial persona Seq2seq models Li et al. (2016b) as well as to the adversarial $hredGAN$ (Olabiyi et al., 2018) with no explicit persona.

## 4.1 Datasets

**TV Series Transcripts** dataset (Serban et al., 2016). We train all models on transcripts from the two popular TV drama series, Big Bang Theory and Friends. Following a similar preprocessing setup in Li et al. (2016b), we collect utterances from the top 12 speakers from both series to construct a corpus of 5,008 lines of multi-turn dialogue. We split the corpus into training, development, and test set with a 94%, 3%, and 3% proportions, respectively, and pair each set with a corresponding attribute file that maps speaker IDs to utterances in the combined dataset.

Due to the small size of the combined transcripts dataset, we first train our model on the larger Movie Triplets Corpus (MTC) by Banchs (2012) which consists of 240,000 dialogue triples. We pre-train

Table 1: $phredGAN$ vs. Li et al. (2016b) on BBT Friends TV Transcripts.

| Model | Teacher Forcing Perplexity | Autoregression | | | | Human Evaluation |
| | | BLEU | ROUGE-2 | DISTINCT-1/2 | NASL | |
|---|---|---|---|---|---|---|
| **TV Series** | | | | | | |
| SM | **22.13** | 1.76 % | 22.4 % | 2.50%/18.95% | 0.786 | 0.5566 |
| SAM | 23.06 | 1.86 % | 20.52 % | 2.56%/18.91% | 0.689 | 0.5427 |
| $hredGAN$ | 28.15 | 2.14 % | 6.81 % | 1.85 %/6.93 % | 1.135 | 0.5078 |
| $phred$ | 30.94 | 2.41 % | 14.03 % | 0.66 %/2.54 % | 1.216 | 0.3663 |
| $phredGAN_a$ | 25.10 | **3.07 %** | **30.47 %** | **2.19 %/19.02 %** | **1.218** | **0.6127** |
| $phredGAN_d$ | 28.19 | 2.76 % | 14.68 % | 0.70 %/4.76 % | 1.163 | 0.4284 |

Table 2: $phredGAN$ vs. Li et al. (2016b) on UDC.

| Model | Teacher Forcing Perplexity | Autoregression | | | | Human Evaluation |
| | | BLEU-2/4 | ROUGE-2 | DISTINCT-1/2 | NASL | |
|---|---|---|---|---|---|---|
| **UDC** | | | | | | |
| SM | 28.32 | 0.437%/∼ 0% | 9.19 % | 1.61%/5.79% | 0.506 | 0.4170 |
| SAM | **26.12** | 0.490%/∼ 0% | 10.23 % | 1.85%/6.85% | 0.512 | 0.4629 |
| $hredGAN$ | 48.18 | **2.16%**/∼ 0% | 11.68 % | **5.16%/18.21%** | 1.098 | **0.5876** |
| $phred$ | 34.67 | 0.16%/∼ 0% | 7.41% | 0.56%/1.44% | 0.397 | 0.4718 |
| $phredGAN_a$ | 31.25 | 1.94%/∼ 0% | **19.15%** | 1.05%/5.28% | **1.520** | 0.4558 |
| $phredGAN_d$ | 28.74 | 2.02%/**0.10%** | 16.82% | 1.38%/5.77% | 1.387 | **0.5817** |

our model on this dataset to initialize our model parameters to avoid overfitting on a relatively small persona TV series dataset. After pre-training on MTC, we reinitialize the attribute embeddings in the generator from a uniform distribution following a Xavier initialization (Glorot & Bengio, 2010) for training on the combined person TV series dataset.

**Ubuntu Dialogue Corpus** (UDC) dataset (Serban et al., 2017b). We train our model on 1.85 million conversations of multi-turn dialogue from the Ubuntu community hub, with an average of 5 utterances per conversation. We assign two types of speaker IDs to utterances in this dataset: questioner and helper. We follow a similar training, development, and test split as the UDC dataset in Olabiyi et al. (2018), with 90%, 5%, and 5% proportions, respectively, and pair each set with a corresponding attribute file that maps speaker IDs to utterances in the combined dataset

While the overwhelming majority of utterances in UDC follow two speaker types, the dataset does include utterances that do not classify under either a questioner or helper speaker type. In order to remain consistent, we assume that there are only two speaker types within this dataset and that the first utterance of every dialogue is from a questioner. This simplifying assumption does introduce a degree of noise into each persona model's ability to construct attribute embeddings. However, our experiment results demonstrate that both $phredGAN_a$ and $phredGAN_d$ is still able to differentiate between the larger two speaker types in the dataset.

## 4.2 Evaluation Metrics

We use similar evaluation metrics as in Olabiyi et al. (2018) including perplexity, BLEU (Papineni et al., 2002), ROUGE (Lin, 2014), distinct n-gram (Li et al., 2016a) and normalized average sequence length (NASL) scores. For human evaluation, we follow a similar setup as Li et al. (2016a), employing crowd-sourced judges to evaluate a random selection of 200 samples. We present both the multi-turn context and the generated responses from the models to 3 judges and asked them to rank the general response quality in terms of relevance, informativeness, and persona. For $N$ models, the model with the lowest quality is assigned a score 0 and the highest is assigned a score N-1. Ties are not allowed. The scores are normalized between 0 and 1 and averaged over the total number of samples and judges.

Table 3: $hredGAN$ vs $phredGAN$ vs Li et al. (2016b) on TV Transcripts

| Model | Speaker | Response |
|---|---|---|
| **Big Bang Theory TV Series** | | |
| Context_0 | Sheldon | Do you like me? |
| $phredGAN_a$ | Leonard | Yes ? |
| $phredGAN_d$ | Leonard | You can ' t do that ? ! |
| $phredGAN_a$ | Penny | No , you don ' t understand . You ' re an idiot . |
| $phredGAN_d$ | Penny | You ' re right . . . |
| $phredGAN_a$ | Howard | Well , I don ' t know what to say . This is the part I ' ve ever seen . |
| $phredGAN_d$ | Howard | You ' re right , you ' re just gonna go with this guy . |
| $phredGAN_a$ | Raj | Well , you know , we could be a little more than my friends ' s friends . |
| $phredGAN_d$ | Raj | You know , I can ' t can ' t do I can ' go , I can ' t do that . . . |
| SM | Leonard/Penny/Howard/Raj | Yeah , I ' m sure . Oh , but I can ' t believe it . I ' m not gonna be in here |
| SAM | Leonard/Penny/Howard/Raj | I ' m not going to be a good time |
| $hredGAN$ | NA | Okay , I ' m gonna go see my friend . |
| Context_0 | Howard | We can't go home. |
| Context_1 | Leonard | Alright, I'll be right back. |
| $phredGAN_a$ | Howard | Okay , well , before you and Penny get out and get married with Penny just get married. |
| $phredGAN_d$ | Howard | You ' ll see , I ' ll just go . |
| SM | Howard | Okay . |
| SAM | Howard | Okay , well , you ' re really good at this . |
| $hredGAN$ | NA | Okay , I ' m gonna go see my old . |
| Context_0 | Amy | Spent a lot of my childhood throwing coins into wishing wells hoping for friends. |
| Context_1 | Penny | Whoa, whoa, give me a second to catch up here. Hold on. |
| Context_2 | Amy | That'd be my boyfriend. Happier playing his dopey Star Trek game with his friends than hanging out with me. |
| $phredGAN_a$ | Penny | You know , when you ' re being so nice about Sheldon and I want to take him to like his own girlfriend |
| $phredGAN_d$ | Penny | You , I ' m sorry , I didn ' t have it . d be back in the bathroom . . |
| SM | Penny | I ' m sorry , I ' m sorry . I ' m just saying that you ' re going to have to answer to |
| SAM | Penny | It ' s not gonna happen . |
| $hredGAN$ | NA | Okay , I ' m gonna go see my new . |
| Context_0 | Sheldon | I recently read that during World War Two, Joseph Stalin had a research program to create supersoldiers by having women impregnated by gorillas. |
| $phredGAN_a$ | Howard | You ' re right . I don ' t know what you ' re talking about , Sheldon . |
| $phredGAN_d$ | Howard | You ' re kidding . . |
| SM | Howard | Well , my last was a little . I was just trying to be a good time . |
| SAM | Howard | That would be great . |
| $hredGAN$ | NA | Okay , I ' m gonna go see my new friend , and then we ' ll go . |
| **Friends TV Series** | | |
| Context_0 | Rachel | Do you like me? |
| $phredGAN_a$ | Joey | Yeah . |
| $phredGAN_d$ | Joey | You ' re right . . . . |
| $phredGAN_a$ | Ross | Well , no , no , no . I mean , I just , I don ' t know . . . |
| $phredGAN_d$ | Ross | You ' want to know sex . . |
| $phredGAN_a$ | Monica | Well , I do . And I ' m not gonna tell you another guy . |
| $phredGAN_d$ | Monica | You ' re a , you ' re really good . . . . |
| $phredGAN_a$ | Chandler | No , I do not . You are a mean little man . . |
| $phredGAN_d$ | Chandler | You ' re right , I can ' t do that . |
| SM | Joey/Ross/Monica/Chandler | Yeah , I ' m sure . Oh , but I can ' t believe it . I ' m not gonna be in here |
| SAM | Joey/Ross/Monica/Chandler | I ' m not going to be a good time . |
| $hredGAN$ | NA | Okay , I ' m gonna go see my friend . |

## 4.3 BASELINE

We compare the non-adversarial persona HRED model, PHRED with the adversarially trained ones, i.e. $hredGAN$, $phredGAN_a$ and $phredGAN_d$, to demonstrate the impact of adversarial training. Please note that no noise was added to the PHRED model.

We also compare the persona models to Li et al.'s work (Li et al., 2016b) which uses a Seq2Seq framework in conjunction with learnable persona embeddings. Their work explores two persona models in order to incorporate vector representations of speaker interaction and speaker attributes into the decoder of their Seq2Seq model i.e., Speaker model (SM) and Speaker-Addressee model (SAM). All reported results are based on our implementation of their models in Li et al. (2016b).

## 4.4 HYPERPARAMETER SEARCH

For both $phredGAN_a$ and $phredGAN_d$, we determine the noise injection method and the optimum noise variance $\alpha$ that allows for the best performance on both datasets. We find that $phredGAN_d$ performs optimally with word-level noise injection on both Ubuntu and TV transcripts, while $phredGAN_a$ performs the best with utterance-level noise injection on TV transcripts and word-level injection on UDC. For all $phredGAN$ models, we perform a linear search for optimal noise variance values between 1 and 30 at an increment of 1, with a sample size of $L = 1$. For $phredGAN_d$, we obtain an optimal $\alpha$ of 4 and 6 for the UDC and TV Transcripts respectively. For $phredGAN_a$,

we obtain an optimal value of 2 and 5 for the combined TV series dataset and the much larger UDC respectively.

## 4.5 RESULTS

We will now present our assessment of performance comparisons of $phredGAN$ against the baselines, PHRED, $hredGAN$ and Li et al.'s persona Seq2Seq models.

## 4.6 QUANTITATIVE ANALYSIS

We first report the performance on TV series transcripts in table 1. The performance of both SM and SAM models in Li et al. (2016b) compared to the hredGAN shows a strong baseline and indicates that the effect of persona is more important than that of multi-turn and adversarial training for datasets with weak multiple persona. However, once the persona information is added to the $hredGAN$, the resulting $phredGAN$ shows a significant improvement over the SM and SAM baselines with $phredGAN_a$ performing best. We also observe that PHRED performs worse than the baseline S(A)M models on a number of metrics but we attribute this to the effect of persona on a limited dataset that results into less informative responses. This behavior was also reported in Li et al. (2016b) where the persona models produce less informative responses than the non-personal Seq2seq models but it seems to be even worse in multi-turn context. However, unlike the Speaker-Addressee and PHRED models that suffer from lower response quality due to persona conditioning, we note that conditioning the generator and discriminator of $phredGAN$ on speaker embeddings does not compromise the systems ability to produce diverse responses. This problem might have been alleviated by the adversarial training that encourages the generator model to produce longer, more informative, and diverse responses that have high persona relevance even with a limited dataset.

We also compare the models performances on the UDC. The evaluation result is summarized in table 2. While the deleterious effect of persona conditioning on response diversity is still worse with PHRED than with S(A)M models, we note that $hredGAN$ performs much better than the S(A)M models. This is because, the external persona only provides just a little more information than is already available from the UDC utterances. We also note an improvement of $phredGAN$ variants over the $hredGAN$ in a variety of evaluation metrics including perplexity, ROUGE with the exception of distinct n-grams. This is expected as $phredGAN$ should be generally less diverse than $hredGAN$ since the number of distinct data distribution modes is more for $phredGAN$ dataset due to the persona attributes. However, this leads to better response quality with persona, something not achievable with $hredGAN$. Also, the much better ROUGE(F1) score indicates that $phredGAN$ is able to strike a better balance between diversity and precision while still capturing the characteristics of the speaker attribute modality in the UDC dataset. Within the $phredGAN$ variants, $phredGAN_d$ seems to perform better. This is not surprising as speaker classification is much easier on UDC than on TV series. The attribute discriminator, $D_{att}$ is able to provide more informative feedback on UDC than on TV series where it is more difficult to accurately predict the speaker. Therefore, we recommend $phredGAN_a$ for datasets with weak attribute distinction and $phredGAN_d$ for strong attribute distinction.

## 4.7 QUALITATIVE ANALYSIS

In addition to the quantitative analysis above, we report the results of the human evaluation in the last column of tables 1 and 2 for the TV Series and UDC datasets respectively. The human evaluation scores largely agrees with the automatic evaluations on the TV Series with $phredGAN_a$ clearly giving the best performance. However, on the UDC, both $hredGAN$ and $phredGAN_d$ performs similarly which indicates that there is a trade off between diversity and persona by each model. We believe this is due to the strong persona information that already exists in the UDC utterances.

An additional qualitative assessment of these results are in Table 3 with responses from several characters in the TV series dataset and the two characters in UDC.

We see that for TV drama series, $phredGAN$ responses are comparatively more informative than that of the Speaker-Addressee model of Li et al. (2016b). For example, all the characters in the TV series respond the same to the dialogue context. Similar behavior is reported in Li et al. (2016b) where for the Speaker-Addressee model, nearly all the characters in the TV series respond with

Table 4: $hredGAN$ vs $phredGAN$ vs Li et al. (2016b) on UDC

| Model | Speaker | Response |
|-------|---------|----------|
| **UDC** | | |
| Context_0 | asker | all i need some quick amarok help. i cant hear my music |
| Context_1 | helper | is amarok muted? |
| Context_2 | asker | no |
| $phredGAN_a$ | helper | use the UNK drivers , and then run the UNK command to get the UNK |
| $phredGAN_d$ | helper | ok , so you  re not using the right driver for the network card , you  re using the UNK ? |
| $phred$ | helper | you can try to install the _UNK package |
| SM | helper | ok , thanks |
| SAM | helper | ok , thanks |
| $hredGAN$ | helper | I have no idea why it would be a bit of a bit. |
| Context_0 | asker | anyone had problems with the kernel update from today? giving me a kernel panic |
| Context_1 | helper | you can select previous kernels at the bootloader (grub) menu on booth. |
| $phredGAN_a$ | asker | it says that it is not installed . . . |
| $phredGAN_d$ | asker | ok , so i  ll have to reinstall the new kernel , i  ll try that. |
| $phred$ | asker | you can try to install the drivers from the live cd |
| SM | asker | I ' m not sure what you mean . . . |
| SAM | asker | I ' m not sure how to do that . . . |
| $hredGAN$ | asker | I ' m not sure how to do that , but I can ' t boot from a CD . . . |
| Context_0 | asker | how do I install Ubuntu? |
| $phredGAN_a$ | helper | use the alternate cd , it should be in the repos , its a good place to get the source of the kernel |
| $phredGAN_d$ | helper | ok , so you have to reinstall the kernel from the CD , and you can  t install the iso to the CD |
| $phred$ | helper | yes |
| SM | helper | you can use the command line |
| SAM | helper | what is your question ? |
| $hredGAN$ | helper | you can use the _UNK to install the _UNK |

"Of course I love you." to the dialogue context, "Do you love me?" despite the fact that some of the responders sometimes have unfriendly relationship with the addressee. Many of the novel situations explored by $phredGAN$ are unachievable with the Speaker-Addressee model due to lack of informative responses. For example, by conditioning as Sheldon from The Big Bang Theory and asking "Do you like me?", our model responds with annoyance if conditioned as Penny ("No, you don't understand. You're an idiot"), brevity with Leonard ("Yes?") and sarcasm with Raj ("Well , you know , we could be a little more than my friend's friends.") The wide range of responses indicate our model's ability to construct distinct attribute embeddings for each character even from a limited dataset. The other interesting responses in table 3 indicate $phredGAN$'s ability to infer not only the context of the conversation but important character information about the addressee.

We also see similar results with our model's output on UDC in table 4. We demonstrate that by conditioning as either a helper or questioner from the UDC dataset, $phredGAN$ models are able to respond differently to input utterances as well as stay close to the context of the conversation.

## 5 CONCLUSION AND FUTURE WORK

In this paper, we improve upon state-of-the-art persona-based response generation models by exploring two persona conversational models: $phredGAN_a$ which passes the attribute representation as an additional input into a traditional adversarial discriminator, and $phredGAN_d$ a dual discriminator system which in addition to the adversarial discriminator from $hredGAN$, collaboratively predicts the attribute(s) that are intrinsic to the input utterance. Both systems demonstrate quantitative improvements upon state-of-the-art persona conversational systems such as the work from Li et al. (2016b) with respect to both quantitative automatic and qualitative human measures.

Our analysis also demonstrates how both variants of $phredGAN$ perform differently on datasets with weak and strong modality. One of our future direction is to take advantage of $phredGAN_d$'s ability to predict utterance attribute such as speaker identity from just the utterance. We believe its performance can be improved even with weak modality by further conditioning adversarial updates on both the attribute and adversarial discriminator accuracies. Overall, this paper demonstrates clear benefits from adversarial training of persona generative dialogue system and leaves the door open for more interesting work to be accomplished in this domain.

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

APPENDIX

---

**Algorithm 1** Adversarial Learning of phredGAN

---

**Require:** A generator $G$ with parameters $\theta_G$.
**Require:** An adversarial discriminator $D_{adv}$ with parameters $\theta_{D_{adv}}$.
**Require:** An attribute discriminator $D_{att}$ with parameters $\theta_{D_{att}}$.
**Require:** Training hyperparameters, $isTarget$, $\lambda_{G_{att}}$, $\lambda_{G_{adv}}$, and $\lambda_M$.
  **for** number of training iterations **do**
    Initialize $cRNN$ to zero_state, $\boldsymbol{h_0}$
    Sample a mini-batch of conversations, $\boldsymbol{X} = \{X_i, C_i\}_{i=1}^N$, $\boldsymbol{X_i} = \big( (X_1, C_1), (X_2, C_2), \cdots, (X_i, C_i) \big)$ with $N$ utterances.
    Each utterance mini batch $i$ contains $M_i$ word tokens.
    **for** $i = 1$ **to** $N - 1$ **do**
      Update the context state.
      $\boldsymbol{h_i} = cRNN(eRNN(E(X_i)), \boldsymbol{h_{i-1}}, C_i)$
      Compute the generator output similar to Eq. (11) in Olabiyi et al. (2018).
      $P_{\theta_G}\big( Y_i |, Z_i, \boldsymbol{X_i}, C_{i+1} \big) = \big\{ P_{\theta_G}\big( Y_i^j | X_{i+1}^{1:j-1}, Z_i^j, \boldsymbol{X_i}, C_{i+1} \big) \big\}_{j=1}^{M_{i+1}}$
      Sample a corresponding mini batch of utterance $Y_i$.
      $Y_i \sim P_{\theta_G}\big( Y_i |, Z_i, \boldsymbol{X_i}, C_{i+1} \big)$
    **end for**
    Compute the adversarial discriminator accuracy $D_{adv}^{acc}$ over $N-1$ utterances $\{Y_i\}_{i=1}^{N-1}$ and $\{X_{i+1}\}_{i=1}^{N-1}$
    **if** $D_{adv}^{acc} < acc_{D_{adv}^{th}}$ **then**
      **if** $isTarget$ **then**
        Update $phredGAN_d$'s $\theta_{D_{adv}}$ and $\theta_{D_{att}}$.
        $\sum_i [\nabla_{\theta_{D_{adv}}} \log D_{adv}(\boldsymbol{h_i}, X_{i+1}) + \nabla_{\theta_{D_{adv}}} \log \big( 1 - D_{adv}(\boldsymbol{h_i}, Y_i) \big) + \nabla_{\theta_{D_{att}}} - \log D_{att}(C_{i+1} | \boldsymbol{h_i}, X_{i+1})]$
      **else**
        Update $phredGAN_a$'s $\theta_{D_{adv}}$ with gradient of the discriminator loss.
        $\sum_i [\nabla_{\theta_{D_{adv}}} \log D_{adv}(\boldsymbol{h_i}, C_{i+1}, X_{i+1}) + \nabla_{\theta_{D_{adv}}} \log \big( 1 - D_{adv}(\boldsymbol{h_i}, C_{i+1}, Y_i) \big)]$
      **end if**
    **end if**
    **if** $D_{adv}{}^{acc} < acc_{G_{th}}$ **then**
      Update $\theta_G$ with the generator's MLE loss only.
      $\sum_i [\nabla_{\theta_G} - \log P_{\theta_G}\big( Y_i |, Z_i, \boldsymbol{X_i}, C_{i+1} \big)]$
    **else**
      Update $\theta_G$ with attribute, adversarial and MLE losses.
      $\sum_i [\lambda_{G_{att}} \nabla_{\theta_G} - \log D_{att}(C_{i+1} | \boldsymbol{h_i}, Y_i) \qquad + \qquad \lambda_{G_{adv}} \nabla_{\theta_G} \log D_{adv}(\boldsymbol{h_i}, C_{i+1}, Y_i) \qquad +$
      $\lambda_M \nabla_{\theta_G} - \log P_{\theta_G}\big( Y_i |, Z_i, \boldsymbol{X_i}, C_{i+1} \big)]$
    **end if**
  **end for**

---

# 6 RESULTS - DISCRIMINATOR

After training both $phredGAN$ models on the TV series and UDC datasets, we ran inference on some example dialogue contexts. The responses and their discriminator scores from $phredGANs$ are listed in Tables 6, and 7. The tables shows that $phredGAN$ (i) can handle multi-turn dialogue context with utterances and corresponding persona attributes; (ii) generates responses conditioned on a persona attribute; (iii) generates multiple responses per dialogue context and score their human likelihood by the discriminator; and (iv) in case of $phredGAN_d$, can predict the attribute such as speaker identity that might have produced the utterance. We observe that the discriminator score(s) is/are generally reasonable with longer, more informative and more persona-related responses receiving higher scores. It worth to note that this behavior, although similar to the behavior of a human judge is learned without supervision. More so, we observe that $phredGAN$ responses retain contextual consistency sometimes referencing background information that is inherent in the conversation between two speakers. For example, in the second sample of the TV series in Table 6, $phredGAN_a$ generator, conditioned on Leonard refers to Sheldon by name who is the second interlocutor. Also, in the third sample, $phredGAN_a$, conditioned on Raj refers to Penny when responding to Leonard who happens to be Penny's boy friend. We see similar persona-based response generation for the UDC dataset with distinct communication style between the asker and the helper. For example, in Table 7, when the asker could not hear some music, $phredGAN_d$, conditioned on helper suggested the asker might not be using the right driver. For the purpose of completion, we also show some samples from PHRED generator on both UDC and TV series dataset in Table 5.

Table 5: Sample of PHRED outputs on UDC and TV Series

| Source | Speaker | Utterance |
|---|---|---|
| **Big Bang Theory and Friends TV Series** | | |
| Context_0 | Howard | Okay, you have to understand something, we're in a hospital right now. |
| Response_0 | Penny | Oh , I ' m sorry . I was just trying to be a girl . |
| Context_0 | Sheldon | Did I? Well, that can't be right. No one's ever done that before. Except me, because I just did it. Sheldon and his brain, yeah |
| Response_0 | Leonard | I ' m sorry . I just don ' t want to be a man who ' s going to be my girlfriend . |
| Context_0 | Leonard | She's gotten really hooked on Age of Conan, she's playing non-stop. |
| Response_0 | Raj | Oh , I ' m sorry . I ' m sorry . I ' m sorry . I ' m sorry . I ' m |
| **UDC** | | |
| Context_0 | asker | all i need some quick amarok help. i cant hear my music |
| Context_1 | helper | is amarok muted? |
| Context_2 | asker | no |
| Response_0 | helper | you can try to install the _UNK package |
| Context_0 | asker | anyone had problems with the kernel update from today? giving me a kernel panic |
| Context_1 | helper | you can select previous kernels at the bootloader (grub) menu on booth. |
| Response_0 | asker | you can try to install the drivers from the live cd |
| Context_0 | asker | how do I install Ubuntu? |
| Response_0 | helper | yes |

Table 6: Ranked $phredGAN_a$ outputs according to adversarial word-level discrimination score

| Source | Speaker | $D_{adv}(G(.))$ | Utterance |
|---|---|---|---|
| **Big Bang Theory and Friends TV Series** | | | |
| Context_0 | Howard | NA | Okay, you have to understand something, we're in a hospital right now. |
| Response_0 | Penny | 0.1705 | Okay, I'm sorry. I'm not really even like that. |
| Response_1 | Penny | 0.1426 | Okay, I'm sorry, I'm sorry. |
| Response_2 | Penny | 0.0688 | Okay. |
| Context_0 | Sheldon | NA | Did I? Well, that can't be right. No one's ever done that before. Except me, because I just did it. Sheldon and his brain, yeah |
| Response_0 | Leonard | 0.3976 | Sheldon , what are you doing ? |
| Response_1 | Leonard | 0.3503 | Sheldon ? |
| Context_0 | Leonard | NA | She's gotten really hooked on Age of Conan, she's playing non-stop. |
| Response_0 | Raj | 0.4890 | Okay , so we know , what about you and Penny doing here ? |
| Response_1 | Raj | 0.3586 | Okay , so we would have to say that about people ? |
| Response_2 | Raj | 0.1113 | Okay , let ' s go . |
| **UDC** | | | |
| Context_0 | asker | NA | all i need some quick amarok help. i cant hear my music |
| Context_1 | helper | NA | is amarok muted? |
| Context_2 | asker | NA | no |
| Response_0 | helper | 0.3079 | use the UNK drivers , and then run the " UNK " command to get the UNK |
| Response_1 | helper | 0.1283 | what is the error message ? |
| Response_2 | helper | 0.0725 | what version of ubuntu ? |
| Context_0 | asker | NA | anyone had problems with the kernel update from today? giving me a kernel panic |
| Context_1 | helper | NA | you can select previous kernels at the bootloader (grub) menu on booth. |
| Response_0 | asker | 0.3665 | it says that it is not installed . . . |
| Response_1 | asker | 0.3195 | i'm not sure what i can find . . . |
| Response_2 | asker | 0.0186 | it's a UNK , I'm not sure of the way . |
| Context_0 | asker | NA | how do I install Ubuntu? |
| Response_0 | helper | 0.5797 | use the alternate cd , it should be in the repos , it's a good place to get the source of the kernel |
| Response_1 | helper | 0.1984 | use the UNK package , it should work . . . |
| Response_2 | helper | 0.0131 | use the UNK |

Table 7: Sample of $phredGAN_d$ outputs with dual discriminator scores

| Source | Speaker | $D_{adv}(G(.))$ | Prediction | $D_{att}(G(.))$ | Utterance |
|---|---|---|---|---|---|
| **Big Bang Theory TV Series** | | | | | |
| Context_0 | Howard | NA | NA | NA | Okay, you have to understand something, were in a hospital right now. |
| Response_0 | Penny | 0.7389 | Rachel | 0.3101 | You can ' believe me , you can ' t can ' t do it . . |
| Response_1 | Penny | 0.5324 | Rachel | 0.3101 | You ' re right , I'm sorry , I ' go , I can ' t do that . . . |
| Response_2 | Penny | 0.4394 | Rachel | 0.3101 | You ' re right . |
| Context_0 | Sheldon | NA | NA | NA | Did I? Well, that cant be right. No ones ever done that before. Except me,because I just did it. Sheldon and his brain, yeah. |
| Response_0 | Leonard | 0.7028 | Rachel | 0.3805 | You ' re really good . |
| Response_1 | Leonard | 0.5998 | Rachel | 0.3805 | You know , I can ' t believe you can ' to do that . |
| Response_2 | Leonard | 0.4384 | Rachel | 0.3805 | You ' re right . |
| Context_0 | Leonard | NA | NA | NA | Shes gotten really hooked on Age of Conan, shes playing non-stop. |
| Response_0 | Raj | 0.6707 | Rachel | 0.2867 | You know , I , I just want to be friends with you . |
| Response_1 | Raj | 0.6451 | Rachel | 0.2867 | You ' want on to do that ? |
| Response_2 | Raj | 0.4138 | Rachel | 0.2867 | You okay ? |
| **UDC** | | | | | |
| Context_0 | asker | NA | NA | NA | all i need some quick amarok help. i cant hear my music |
| Context_1 | helper | NA | NA | NA | is amarok muted? |
| Context_2 | asker | NA | NA | NA | no |
| Response_0 | helper | 0.4489 | helper | 0.8582 | ok , so you ' re not using the right driver for the network card , you ' re using the _UNK ? |
| Response_1 | helper | 0.2312 | helper | 0.8582 | ok , so you ' re not using the right driver , right ? |
| Response_2 | helper | 0.1381 | helper | 0.8582 | ok , so you are you using the _UNK ? |
| Context_0 | asker | NA | NA | NA | anyone had problems with the kernel update from today? giving me a kernel panic |
| Context_1 | helper | NA | NA | NA | you can select previous kernels at the bootloader (grub) menu on booth. |
| Response_0 | asker | 0.3431 | asker | 0.8126 | ok , so i ' ll have to reinstall the new kernel , i ' ll try that . |
| Response_1 | asker | 0.2518 | asker | 0.8126 | ok , so i ' just reinstall the ubuntu version ? |
| Response_2 | asker | 0.1005 | asker | 0.8126 | ok , thanks :) |
| Context_0 | asker | NA | NA | NA | how do I install Ubuntu? |
| Response_0 | helper | 0.4306 | helper | 0.8540 | ok , so you have to reinstall the kernel from the CD , and you can ' t install the iso to the CD |
| Response_1 | helper | 0.3783 | helper | 0.8540 | ok , so you have to go to the ubuntu site and see if you have the same version ? |
| Response_2 | helper | 0.1618 | helper | 0.8540 | ok , so are you using the ubuntu version ? ? |

