# OpenReview forum: "An Adversarial Learning Framework for a Persona-based Multi-turn Dialogue Model"
_ICLR.cc/2019/Conference_

### Official Review · AnonReviewer3 · 2018-11-01
**novelty limited and experiments not convincing enough**

**Rating:** 5
**Confidence:** 4

**Review:**

===============================
I have read the authors' response and other reviewers' comments carefully. Thank you for taking great efforts to improve the paper, including providing additional results on human evaluation. (Btw, Table 1 and Table 2 are also much nicer now.)

However, from the reviews it seems that all the reviewers agree that the novelty of this paper is limited, and the contribution is incremental.  I understand that this paper is the first and only work using adversarial framework for persona multi-turn conversation models. However, from the modeling perspective, I still think the novelty is limited.

As a summary, I have updated the score from 4 to 5 to reflect the efforts that the authors have been taken to improve the paper. However, due to reasons above, I still prefer a rejection recommendation.

===============================

Contributions:

The main contribution of this paper is the proposed phredGAN, which is a persona-based GAN framework for multi-turn dialogue modeling. Specifically, a persona-based HRED generator is developed, with two different kinds of discriminator design. Experiments are conducted on both the UDC and the TV series transcript datasets.

Weaknesses:

(1) Novelty: I would say the novelty of this paper is rather limited. This paper heavily rely on the previous hredGAN work (Olabiyi et al., 2018), and extends it by injecting attributes into the system, borrowing ideas from the persona-based Seq2seq model (Li et al, 2016b).

phredGAN_a is a straightforward extension of hredGAN, while phredGAN_d further introduces a collaborative discriminator that tries to predict the attribute that generated the input utterance. However, in summary, I think this paper is not novel enough.

(2) Presentation: The paper is generally easy to follow and understand. However, I would say the paper is poorly written, and needs further polishing. For example, Table 1 & 2 are pretty ugly.

(3) Evaluation: Generally, I think the experiments are not convincing and also not well-executed, with detailed comments listed below.

Questions:

(1) In phredGAN_a, as shown in Eqn. (4), the attribute is used as input of the discriminator, while in phredGAN_d, as shown in Eqn. (5) & (6), the attribute is used as the target of the discriminator. My question is: why not use the attribute as both input & output? That is, why not combine (4) & (6), instead of using (5) & (6)? Please clarify this.

(2) In experiments, Section 3.1, the authors mention that the generator and the discriminator use a shared encoder. However, the generator and discriminator has a different role. Since the encoder is shared, then: in one step, we update the encoder to minimize the GAN objective, in the alternative step, we update the encoder again to maximize the GAN objective. So, how to deal with this conflicting role of encoder during the training? Please clarify this.

(3) From Table 2, it seems that it is difficult to see that phredGAN is better than hredGAN. Can you provide some explanations here?

(4) In Table 4, if the responses generated by hredGAN can be provided, that would be better to demonstrate the advantage of phredGAN. How does phredGAN compare with hredGAN qualitatively?

(5) From Table 1 & 2, it seems to me there is no metric that is specifically designed to evaluate whether the model captures the attribute information. Is there a way to quantitatively evaluate this? For example, pretrain an attribute classifier, or use the collaborative discriminator in the phredGAN_d model to measure how the generated response reflect the attribute. If we can observe the performance of phredGAN is better than that of hredGAN, that would be helpful for the paper.

(6) Since the task is challenging, and the automatic metrics designed for this task is not perfect, like other papers, I think human evaluation is essential and desired for this task. However, such human evaluation is lacked in this paper.

---

> ### Author Response · Authors · 2018-11-09
> **Novelty, experiments and human evaluation**
>
> Thank you for your review.
>
> --Novelty: I would say the novelty of this paper is rather limited.
> We would like to draw your attention to the fact that this is the first and only work addressing adversarial framework for persona multi-turn conversation models with the exploration of different discrimination methods across datasets with varying persona strength. We believe this is unique and important for the future research in this domain.
>
> -- Limitation on Evaluation
> We are retraining SM and SAM so that we can evaluate them across all the metrics similar to the phredGAN. We will also include the hredGAN results on TV series for the purpose of completeness. Finally, we are crowdsourcing human judges to evaluate the models so that we can have a more conclusive comparison.
>
> ---Table 1 & 2 are pretty ugly.
> Can you please clarify this comment so that we can fix any issue with the Tables?
>
> --That is, why not combine (4) & (6), instead of using (5) & (6)? Please clarify this.
> There is nothing that prevents learning this way that but we feel that it is unnecessary as phredGAN_a is already sufficient. The phredGAN_d can be seen as a decomposition of phredGAN_a, i.e., the adversarial discriminator of phredGAN_d makes the real/fake determination just like the hredGAN, while the attribute discriminator makes the attribute determination. This independent decomposition allows for better interpretability but also comes with a loss of performance when the attribute is weak as shown in the results in Tables 1 and 2. This tradeoff is what we wanted to investigate.
>
> --So, how to deal with this conflicting role of encoder during the training? Please clarify this.
> The encoder and embedding parameters are not updated during the discriminator update. However, the gradients flow from the discriminator to the generator through the encoder and the embedding during the adversarial generator update.
>
> --From Table 2, it seems that it is difficult to see that phredGAN is better than hredGAN. Can you provide some explanations here?
> We believe that there is a trade-off between response quality and diversity with persona. Similar behavior has been observed by Li et al, 2016b. Therefore, the hredGAN is inherently more diverse than phredGAN but generates less quality responses. Also, from the data distribution perspective, the phredGAN separates the original hredGAN response distribution into multiple modalities based on the number of attributes. This allows it to generate more relevant samples but less room for diversity. This agrees with the results in Table 2 as discussed in section 4.6 in the paper.
>
> --In Table 4, if the responses generated by hredGAN can be provided, that would be better to demonstrate the advantage of phredGAN. How does phredGAN compare with hredGAN qualitatively?
> We will add the corresponding samples from the hredGAN to Table 3 to show that the phredGAN is actually better for datasets with persona information albeit with some loss of diversity.
>
> --If we can observe the performance of phredGAN is better than that of hredGAN, that would be helpful for the paper.
> We will capture this in the human evaluation exercise.
>
> In case you have other questions while we populate the new results, please let me know.

---

### Official Review · AnonReviewer1 · 2018-11-02
**The idea seems interesting but both the contribution and evaluation are not strong enough for the paper to be accepted, the proposed objective function is also questionable.**

**Rating:** 4
**Confidence:** 4

**Review:**

This paper proposes an extension to hredGAN, which is an adversarial framework for multi-turn dialogue model, to simultaneously learns a set of attribute embeddings that represents the persona of each speaker and generate persona-based responses. The generator of the proposed system phredGAN is conditioned on both the history utterances and the speakers’ persona by concatenating the utterance encoding with attribute embeddings. For discriminator, the authors explore two versions: 1) phredGAN_a takes attributes as inputs; 2) phredGAN_d adds a dual discriminator that predicts the attribute(s) for each utterance.


Strength: 1) to the best of my knowledge, adding persona information to an adversarial multi-turn dialogue model is novel; 2) the authors explore two different approaches to build the discriminator(s) and the idea of adding a second discriminator that predicts the attributes seems interesting.


Weakness:

1) Novelty: The idea of learning speaker-specific attribute embeddings is very similar to the Speaker Model proposed by Li et al.(2016) http://www.aclweb.org/anthology/P16-1094 and the proposed system only makes minor changes to hredGAN https://arxiv.org/abs/1805.11752.


2) Presentation:
The writing of this paper is a little hard to follow, for example, it presents the two discriminators after the objective function (Equation 2) and does not explain the intuition behind each model. In Equation 2, the objective function, why training the discriminator to minimize the attributes prediction probability？ Simply saying the attribute prediction loss is collaborative is not clear enough. Or is the min for the second term a typo?


3) Model:
The idea of adding a discriminator that predicts the attributes seems interesting. However the loss is not adversarial for the second discriminator (Equation 6), you should not indicate L_att is GAN in your notation. I’m also not convinced that this should be collaborative. Despite that the “discriminator” is trying to predict the correct attribute id, the input of the two terms in Equation 6 is different, one comes from the true data, the other comes from the generator. Shouldn’t the discriminator try to differentiate these two cases? Otherwise, it’s not a discriminator (also raise the question for Equation 2, why argmin min(L_att)).


4) Evaluation：
The evaluation is not strong enough to demonstrate the benefit of the proposed model.

a. It only compares against one previous work that takes speaker identity into account on one dataset. Despite that the authors apply several different metrics to evaluate the proposed model, they only compare with previous models by Li et al. (2016) on perplexity and BLEU.

b. The perplexity scores of the proposed models are worse than SAM by Li et al. (2016). The authors explain this by stating that the entropy for a multi-turn model is supposed to be higher than the single-turn model. It’s better to provide a more rigorous analysis. For a fair comparison, they could also train the proposed model using only one-turn history, which should be identical to Li et al.’s setting (How many turns history are you using?). The improvements of the BLEU score might also be the consequence of substituting the past generated sequence in the generator with ground truth (since the model uses the same training algorithm as hredGAN https://arxiv.org/abs/1805.11752). It’s unclear if this is the cause unless the authors provide the comparison among SAM, hredGAN, and phredGAN.

c. Table 2 compares the non-persona hredGAN with phredGAN on UDC, but the authors do not provide a comparison between these two on the TV dataset in Table 1.

d. The comparison between phredGAN_a and phredGAN_d is inconsistent for the two datasets (Table 1 and Table 2).

---

> ### Author Response · Authors · 2018-11-09
> **Typos in objective function Eq. (6) and human evaluation**
>
> Thank you for your review and for recognizing the contribution of this work.
> -- Limitation on Evaluation
> We are retraining SM and SAM so that we can evaluate it across all the metrics similar to the phredGAN. We will also include the hredGAN results on TV series for the purpose of completeness. Finally, we are crowdsourcing human judges to evaluate the models so that we can have a more conclusive comparison.
> --Minor typos and L_att
> Indeed there is omission of negative signs in front of "log"s in Eq.(6) which are responsible for your questions 2 and 3. Also, We will remove GAN from L_att notation as suggested. The proposed L_att is similar to the idea in Class-conditional Superresolution with GAN (http://cs231n.stanford.edu/reports/2017/pdfs/314.pdf) . The only difference is that we didn’t pre-train the attribute discriminator as they did. The idea is to use the attribute discriminator to transfer the attribute differences learned from the ground truth to the generator output. Hence the reason for the collaborative setup. This improves the model interpretability since the attribute discriminator necessarily reveals the strength of the attribute modality in the ground truth. The performance of the discriminator during training already reveals if there’s any distinct attribute from the ground truth that can be transferred to the generator. Please note that the discriminator is only updated by the ground truth portion (LHS) during training while the generator is updated by the generator output portion (RHS). We will include more explanation in the final version.
>
> In case you have other questions, please don’t hesitate to ask while we prepare the new results.

---

> > ### Comment · AnonReviewer1 · 2018-11-29
> > **More complete evaluation but results are not good enough**
> >
> > I can see what you're trying to do with Eq (6) after the change but if it's related to the conditional GAN and GAN+class, it's better to cite them and discuss the differences/similarities in the paper.
> >
> > With additional experiments, the evaluation seems to be more complete. However, the results in Table 1 and Table 2 do not show consistent improvements of the proposed models.  For example, if we only look at the human evaluation, it's not clear why phredGAN_a has the highest score among hredGAN, phred, phredGAN_a, phredGAN_d for TV datasets but has the lowest for UDC dataset. It's better if we can see more concrete analysis.
> >
> > I increase my score because of more complete evaluation. However, due to the overall limited novelty/improvements and unclear presentation, the paper is still under my expectation for ICLR.

---

> > > ### Author Response · Authors · 2018-11-30
> > > **Mistakenly switched phred and phredGAN_a 's human score on UDC**
> > >
> > > Thank you for your favorable consideration after reviewing the new results and paper update.
> > >
> > > We will cite the GAN+class paper in the final version and explicitly relate phredGAN_a to conditional GAN and phredGAN_d to GAN+class. Note that to the best of our knowledge, no existing literature has investigated and compare the performance of these approaches for multi-turn dialogue response generation. Hence, we believe this work will be a tremendous resource to the research community and will stimulate more research in this area.
> > >
> > > Also, the human evaluation score for phredGAN_a was mistakenly switched for the score for phred in a rush to complete the paper update before the deadline. We will correct this in the final version. Nonetheless, we believe the reason that hredGAN and phredGAN_d perform better than phredGAN_a is most likely due to:
> > > (i) The strong persona information (Helper vs. Questioner) in the UDC utterances already makes hredGAN a very strong baseline. Therefore, the external persona information did not convey much more additional information than already provided in the utterance as we have described already in the updated paper.
> > > (ii) Also, the noisy nature of the attribute information described in Section 4.1 for UDC worsens the performance of phredGAN_a more than that of phredGAN_d. The attribute is a feature into the adversarial discriminator of phredGAN_a and any incorrectly labelled input affects the adversarial feedback. However for phredGAN_d, the attribute discriminator is able to average out the effect of noisy attribute labels to provide quality feedback to the generator. Note that the adversarial discriminator of phredGAN_d is not affected by this noisy attribute labels.
> > >
> > > Still we expect phredGAN_d to perform  better than phredGAN_a on datasets with strong attribute-utterance correlation such as UDC due to the quality of the attribute feedback provided to the generator.
> > >
> > > We shall add these additional explanations to final version of the paper.
> > >
> > > In case you have other questions, please don’t hesitate to ask.

---

### Official Review · AnonReviewer2 · 2018-11-02
**Extension of HREDGan**

**Rating:** 6
**Confidence:** 3

**Review:**

This paper uses the idea from 'A Persona-Based Neural Conversation Model' by Li et al and incrementally applies it to the 'Multi-turn Dialogue Response Generation in an Adversarial Learning Framework' work-in-progress by Olabiyi et al. The paper by Olabiyi uses the idea of adversarial training to the HRED work by Xing et al (Hierarchical Recurrent Attention Network for Response Generation). The paper shows very promising results for controlling the response generation based on input attributes with adversarial training. Compared to the persona based model, this work seems to outperform that model significantly as reported in Table 1 (in terms of Perplexity/Bleu). It would have been great to see the quantitative comparison in terms of other metrics (if the authors could try to reproduce their results). There are other interesting ways to incorporate attribute information into the dialogue model such as reported in the work of Lee et al (SCALABLE SENTIMENT FOR SEQUENCE-TO-SEQUENCE CHATBOT RESPONSE WITH PERFORMANCE ANALYSIS) - since this paper is primarily about personalization of responses - a comparison to some of the methods used in Lee's work would have been very relevant and made the paper much more convincing in terms of core contributions. The model and architecture is pretty convincing but the paper lacks more in-depth analysis, comparison and evaluation of the model.

---

> ### Author Response · Authors · 2018-11-09
> **Human evaluation ongoing...**
>
> We are reproducing Li et al. 's model and we will also report results on human evaluation with detailed analysis and comparison across all models soon.

---

### Author Response · Authors · 2018-11-27
**Baseline re-implemented and human evaluation results added**

Thank you for your reviews. We have conducted human evaluation of the models presented in the paper and added the results with some discussion to the updated paper. To achieve this, we re-implemented the persona models (Speaker and Speaker-Addressee) in Li et al. (2016) and trained them on the presented datasets in order to make a thorough comparison. We have also updated the paper with most of the changes requested by the reviewers. Please endeavor to review them and let us know if you have additional questions or concerns.

---

### Meta-Review · Area_Chair1 · 2018-12-17
**Interesting preliminary ideas on personalizing dialogue responses that require more work**

**Confidence:** 4
**Recommendation:** Reject

**Metareview:**

This work presents extensions of dialogue systems to simultaneously capture speakers' "personas" (in the framing of Li et al's work) and adapt to them. While the ideas are interesting, reviewers note that the incremental contribution compared to previous work is a bit too limited for ICLR's expectation, without being offset by strongly convincing experimental results. Authors are encouraged to incorporated their ideas into future submissions after having combined them with other insights to provide a stronger overall contribution.